# Neurological and Neurocognitive Impairments in Adults with a History of Prenatal Methylmercury Poisoning: Minamata Disease

**DOI:** 10.3390/ijerph20126173

**Published:** 2023-06-19

**Authors:** Takashi Yorifuji, Tomoka Kadowaki, Mariko Yasuda, Yoko Kado

**Affiliations:** 1Department of Epidemiology, Graduate School of Medicine, Dentistry and Pharmaceutical Sciences, Okayama University, 2-5-1 Shikata, Okayama 700-8558, Japan; p5j44x0d@s.okayama-u.ac.jp; 2Center for Clinical Psychology, Kawasaki Medical School Hospital, 577 Matsushima, Okayama 701-0192, Japan; maryco.0425@gmail.com; 3Department of Psychology, Faculty of Letters, Kansai University, 3-3-35, Yamate-cho, Osaka 564-8680, Japan; kado@kansai-u.ac.jp

**Keywords:** environmental pollution, food contamination, methylmercury compounds, minamata disease, neurocognitive evaluations, neurological examinations

## Abstract

Minamata disease, which happened during the 1950s and 1960s in Minamata, Japan, is a well-known case of food poisoning caused by methylmercury-contaminated fish. Although many children were born, in the affected areas, with severe neurological signs after birth (known as congenital Minamata disease (CMD)), few studies have explored the possible effects of low-to-moderate methylmercury exposure in utero, probably at lower levels than in CMD patients, in Minamata. We, therefore, recruited 52 participants in 2020: 10 patients with known CMD; 15 moderately exposed residents; and 27 non-exposed controls. The average umbilical cord methylmercury concentrations were 1.67 parts per million (ppm) for CMD patients and 0.77 ppm for moderately exposed participants. After conducting four neuropsychological tests, we compared the functions among the groups. Compared with the non-exposed controls, both the CMD patients and moderately exposed residents had worse scores in the neuropsychological tests, although the score decline was more severe in the CMD patients. For example, even after adjusting for age and sex, the CMD patients and moderately exposed residents had 16.77 (95% CI: 13.46 to 20.08) and 4.11 (95% CI: 1.43 to 6.78) lower scores in the Montreal Cognitive Assessment, respectively, than the non-exposed controls. The present study indicates that residents of Minamata who experienced low-to-moderate prenatal methylmercury exposure also have neurological or neurocognitive impairments.

## 1. Introduction

Minamata disease, which occurred in Minamata in the 1950s and 1960s, is well known as a food poisoning caused by fish contaminated with methylmercury [1,2]. The first patient was officially notified to the local Public Health Center on 1 May 1956. The source and transmission mode were contaminated fish and shellfish. The etiologic agent, methylmercury, was discharged from a chemical factory into Minamata Bay and the Shiranui Sea from 1932 until 1968; no effective preventive measures were taken during this period [3,4].

Because the target organ of methylmercury is the central nervous system, the residents affected by methylmercury manifested neurological signs, such as paresthesia, visual field constriction, ataxia, hearing difficulties, and dysarthria [5,6]. Furthermore, in the affected areas, many children who were exposed to methylmercury in utero manifested severe neurological signs after birth [7]. Such individuals have what is termed congenital Minamata disease (CMD). 

It has recently been suggested that, although they do not present severe neurological signs, some residents of Minamata who were born during the same period as the CMD patients have cognitive impairments [8,9]. This may be caused by prenatal exposure to methylmercury, but probably at lower levels than those of CMD patients. However, few studies have explored the effects of low-to-moderate in utero methylmercury exposure in Minamata. Indeed, a report from the U.S. National Research Council on the toxicological effects of methylmercury identified this problem, stating that the “identification of cases was undoubtedly incomplete, particularly among individuals who suffered milder forms of CMD… Findings suggest that many children with less severe forms of CMD were undiagnosed” [10].

To date, several epidemiological studies in Minamata have explored this issue. For example, one study reported that CMD patients, as well as other groups with cognitive deficits in Minamata, had higher methylmercury concentrations in their umbilical cords than those of healthy residents [11]. Furthermore, Fujino et al., conducted a cross-sectional study in 1970 to assess the neurological status of adolescents aged 12 to 15 years born between 1955 and 1958 in the exposed areas; there was a higher prevalence of some neurological signs (e.g., abnormal fine motor coordination) and intellectual disability diagnosed by medical doctors among students in the exposed area compared with a reference population [12]. In the same participants, Futatsuka et al., also demonstrated a higher prevalence of fine motor coordination difficulties (as evaluated by several batteries, such as tapping) compared with the reference population [13]. However, no other detailed epidemiological studies in Minamata have explored the possible impacts of low-to-moderate prenatal exposure to methylmercury, probably at lower levels than in CMD patients, using established neuropsychological tests. Moreover, no studies have evaluated the persistence into adulthood of the possible effects of prenatal exposure to methylmercury (i.e., more than half a century after exposure). 

We, therefore, examined the association of prenatal methylmercury exposure with neurological and neurocognitive functions, with a special focus on low-to-moderate exposure to methylmercury in utero. 

## 2. Materials and Methods

### 2.1. Study Area and Participants

We recruited 52 study participants who provided written informed consent for participation in the study during the research period, from August to October 2020. The participants comprised 10 patients with known CMD, 15 moderately exposed residents, and 27 non-exposed controls. 

All 10 CMD patients were certified as having Minamata disease by the Japanese Government and lived with support from their families or caregivers because of their difficulties [14]. To be certified, residents should have a combination of neurological signs with an exposure history and one of the requirements for the exposure history for CMD patients includes that the patients should have an umbilical methylmercury concentration above 1 ppm [15]. The 15 moderately exposed residents included 13 participants from our previous studies [8,9] and two additional participants. We recruited them independently of their certification status; only one of the 15 residents was certified as having Minamata disease by the Japanese Government. None of the 15 residents needed any support for everyday life, although they were all exposed to methylmercury to some extent in utero. All 10 CMD patients and 15 moderately exposed residents were born between 1952 and 1965 and were raised in Minamata City and surrounding areas, in western Japan (Figure 1). Thus, all 10 CMD patients and 15 moderately exposed residents would have been exposed to methylmercury in utero. The extent of the intrauterine exposure may have been higher for the CMD patients than for the moderately exposed residents.

We also included 27 non-exposed controls with similar ages to the exposed participants. These controls were recruited from a clinic in Niihama, Ehime Prefecture (Figure 1). All control participants were born between 1955 and 1965, raised in Niihama City and surrounding areas, and were not raised in areas exposed to methylmercury at similar levels to Minamata. They were local residents around the clinic. To provide an appropriate comparison, medical staff (such as physicians and nurses) were not included as controls.

One nurse (TK) interviewed each participant using the questionnaire that we created. Questions included participants’ drinking and smoking histories, educational attainment, self-rated health, medical history, and current health status (such as the presence of paresthesia, muscle cramps, or clumsiness). If available, we also collected information on previously measured umbilical cord methylmercury concentrations of exposed participants, which can reflect intrauterine exposure [16].

### 2.2. Neuropsychological Tests

We conducted four neuropsychological tests to evaluate the association of prenatal exposure to methylmercury with the participants’ neuropsychological functions. The tests were selected based on previous studies and the possible effect of exposure to methylmercury in utero and we aimed to assess visuospatial construction, executive functions, and fine motor skills. The CMD patients were unable to undergo some of the tests because of their difficulty.

#### 2.2.1. Japanese Version of the Montreal Cognitive Assessment (MOCA-J)

The Montreal Cognitive Assessment is a brief cognitive screening tool for detecting mild cognitive impairment in older people; we used the Japanese version (the MOCA-J). Its reliability and validity are well established [17]. The MOCA-J assesses domains such as attention, concentration, executive function, memory, language, visuoconstructional skills, conceptual thinking, calculations, and orientation. One medical doctor (TY) administered the test. Higher scores reflect better neuropsychological function. 

#### 2.2.2. Four Subtests from the Wechsler Adult Intelligent Scale III (WAIS-III)

The WAIS-III is a test for measuring intelligence in adults. It contains 14 subtests, of which we selected four: the digit symbol coding, block design, information, and symbol search subtests. Two neuropsychologists (YK and MY) performed the tests. For the digits symbol coding subtest, participants were asked to write symbols that matched numbers by referring to a table of numbers with symbols. This subtest evaluates processing speed, visual short-term memory, visual scanning, and visual-motor coordination. For the block design subtest, participants were asked to arrange blocks to match a design that was formed by the examiner or shown on cards. This subtest assesses spatial problem-solving and manipulative abilities. For the information subtest, participants were asked general knowledge questions to evaluate their ability to acquire, retain, and retrieve information. For the symbol search subtest, participants were asked to determine if the presented group had the same symbols as the sample group. This subtest evaluates processing speed, speed of visual perception, and visual discrimination. We then calculated the standardized scores of the components. Higher scores reflect better neuropsychological function.

#### 2.2.3. Keio Version of the Wisconsin Card Sorting Test (KWCST)

We applied a modified version of the Wisconsin Card Sorting Test (WCST): the KWCST. The WCST is devised to study “abstract reasoning” and “shift of set” [18,19], requiring conceptual formation and cognitive flexibility, and examines executive functions comprehensively. The KWCST was created in Japan and uses fewer response cards (48 cards) than the original WCST [20]. Similar to the WCST, the KWCST measures executive function and can reflect the function of the frontal lobe region of the brain [20]. The clinical utility and validity of this test for assessing executive function has been confirmed [21]. Two neuropsychologists (YK and MY) conducted the test. Participants were directed to sort the cards according to one of three attributes: shape, color, or number. Participants were only told cue words, namely “right” or “wrong”, after each trial, and the sorting criterion were changed without notification. We performed this test in two steps and provided a brief instruction as a hint between the steps. We administered the second step when the score of the participants did not meet the achievement criteria. 

From the KWCST, we used the scores for the following four indices in the first step: categories achieved (CA), perseverative errors of Nelson (PEN), total errors (TE), and non-perseverative errors of Nelson (NPEN). Each measures the following [22]: CA, the number of categories completed during the test (i.e., the sequences of six correct matches); PEN, the persistence of incorrect responses; TE, the total number of incorrect responses; and NPEN, the number of errors obtained by subtracting the TE from the PEN. The scores were standardized by comparing samples in the same age category [23]. Higher CA scores reflect better function, whereas higher PEN, TE, and NPEN scores reflect worse function.

#### 2.2.4. Grooved Pegboard Test

The Grooved Pegboard Test is a dexterity test that measures complex visual-motor coordination. We used the Grooved Pegboard (Model 32025, Lafayette Instrument Company, Lafayette, IN, USA), which consists of 25 holes with randomly positioned slots. One medical doctor (TY) administered the test. Participants were asked to rotate the pegs, which had a key along one side, to match the holes. The test was performed with both the dominant and non-dominant hands. We recorded the duration (in seconds) required to perform each trial, which began when the participants started the task and ended when the last peg was correctly placed. Longer durations reflect worse neuropsychological function.

### 2.3. Statistical Analyses

We compared the demographic characteristics among the three exposure groups (i.e., CMD patients, moderately exposed residents, and non-exposed controls). We then compared the neuropsychological outcomes among the groups using analysis of variance for three-group comparisons or the t-test for two-group comparisons. Multiple linear regression analysis was performed using the non-exposed controls as a reference, adjusted for age (continuous) and sex (binary). However, we did not adjust for parameters that were considered intermediate variables between methylmercury exposure and the outcomes (e.g., educational attainment or self-rated health). For the Grooved Pegboard Test results, we compared the outcomes for dominant and non-dominant hands.

All confidence intervals were calculated at the 95% level. Stata SE version 17 (StataCorp, College Station, TX, USA) was used for all analyses. This study was approved by the Ethics Committee at Okayama University Graduate School of Medicine, Dentistry and Pharmaceutical Sciences and Okayama University Hospital (No. 1912-026). All participants provided written informed consent.

## 3. Results

The demographic characteristics grouped by exposure status are shown in Table 1. The non-exposed group was slightly younger, had higher educational attainment, and had better self-rated health than the exposed groups. Umbilical cord methylmercury concentrations were available from two CMD patients and seven moderately exposed participants; the average concentrations were 1.67 parts per million (ppm) and 0.77 ppm, respectively.

Table 2 shows the distributions of the neuropsychological tests grouped by exposure status. Although the CMD patients were unable to undergo two of the WAIS-III subtests for processing speed and the KWCST, they had the lowest scores among the three groups. Furthermore, the moderately exposed residents had lower scores than the non-exposed controls. 

The results from the multiple linear regression are shown in Table 3. Even after adjusting for age and sex, compared with the non-exposed controls, both the CMD patients and the moderately exposed residents had worse scores on the neuropsychological tests and the score decline was more severe in the CMD patients. For example, the CMD patients and moderately exposed residents had 16.77 (95% confidence interval: 13.46 to 20.08) and 4.11 (95% confidence interval: 1.43 to 6.78) lower MOCA-J scores, respectively, than the non-exposed controls.

## 4. Discussion

In this study, we evaluated the association of prenatal methylmercury exposure with neurological and neurocognitive functions, focusing on low-to-moderate exposure to methylmercury in utero. Compared with the non-exposed controls, both the CMD patients and the moderately exposed residents had worse scores on the neuropsychological tests for fine motor skills, processing speed, visuospatial construction, and executive functions; however, the score decline was more severe in the CMD patients. Moreover, this decline persisted into adulthood, more than half a century after the exposure.

Although the CMD patients were unable to complete all the tests because of their difficulty, the observed neurological and neurocognitive impairments were expected based on previous findings that CMD patients have intellectual disability because of methylmercury exposure [1]. Indeed, a previous review noted a high prevalence of intellectual disability among CMD patients [1]. Moreover, the finding of worse scores in the moderately exposed residents than in the non-exposed controls was also consistent with previous studies in Minamata [8,9,11,12,13]. By examining neurological and neurocognitive functions using established neuropsychological tests, the present findings strengthen the results from previous studies. 

In the present study, the observed neurocognitive impairments, as a result of prenatal methylmercury exposure, are consistent with findings from previous studies that examined the possible effects of methylmercury exposure in utero (at much lower levels than in Minamata) on neurocognitive function [24,25,26]. For example, a systematic review included 27 epidemiologic studies and examined possible developmental neurotoxicity of methylmercury [26]. Most of the included studies were based on communities with high fish intake, while a small number of studies were based on general populations. Even the residents in the communities with high fish intake are considered to have lower methylmercury concentrations (e.g., average maternal hair mercury concentrations <10 ppm) than the exposed population in Minamata (e.g., the median hair mercury concentration among healthy fisherman in Minamata was 30 ppm in 1960 [6]). Then, the review noted that most of the studies found neurodevelopmental impairments because of prenatal methylmercury exposure and the affected neurocognitive domains included areas related with attention, language, motor, and visuospatial functions [26]. The current findings support these previous results.

In the present study, based on our previous studies in Minamata [6,7], we focused on fine motor skills, processing speed, visuospatial construction, and executive functions for the neurological and neurocognitive assessments. The MOCA-J assesses cognitive function in a wide range of domains; the four subtests from the WAIS-III assess processing speed, visual perception, and other functions; the KWCST measures executive functions such as working memory and response inhibition; and the Grooved Pegboard Test assesses visual-motor coordination. Both the CMD patients and the moderately exposed residents had worse scores in these tests than the non-exposed controls. These neuropsychological tests are related to various regions of the brain, such as the frontal and parietal regions, which indicates that the exposed participants had diffuse damage throughout the brain. 

Methylmercury is neurotoxic for the central nervous system and is considered to affect different regions of the brain depending on the exposure timing. When residents in Minamata were exposed to methylmercury after birth, they underwent localized damage in the brain; specifically, the cerebral cortex (somatosensory, primary visual, and primary auditory areas) as well as the cerebellum were damaged [5]. This pathology explains the well-known neurological signs of methylmercury neurotoxicity, including paresthesia, ataxia, visual field constriction, dysarthria, and hearing difficulties. By contrast, CMD patients have been reported to have more diffuse damage in the brain [27]. Therefore, even low-to-moderate prenatal exposure may cause diffuse damage in the brain, which might explain the present findings.

The average methylmercury concentrations among the participants were 1.67 ppm in the CMD patients and 0.77 ppm in the moderately exposed participants. An average of 0.77 ppm in the umbilical cord relates to 19.4 ppm of mercury in maternal hair using the formula proposed by Akagi et al. [28]. As mentioned, on the basis of previous studies included in the systematic review [26], this level is higher than the value that is thought to induce neurocognitive impairments. It is, therefore, reasonable to expect some neurological and neurocognitive deficits even in moderately exposed residents.

The strength of the present study is that we were able to evaluate the effects of prenatal methylmercury exposure in Minamata, where the most severe environmental methylmercury exposure occurred. Moreover, we were able to evaluate the persistence in adulthood of any possible effects from prenatal methylmercury exposure. Nonetheless, the present study also has several limitations. First, the investigators knew the exposure status of the participants, which may have led to outcome misclassification. However, we used established batteries for the neuropsychological assessments, so we expect that such misclassification is likely to be minimal. Second, the exposed participants continued to eat contaminated fish after birth. We were, thus, unable to examine whether prenatal or postnatal exposure was most related to impairments in the neuropsychological tests. Third, we did not adjust for educational attainment or self-rated health because these variables were considered to be intermediate variables between methylmercury exposure and the outcomes, and we were unable to adjust for the socioeconomic status or intelligence quotient of the participants’ parents, which might confound any associations. Moreover, there is a possibility of residual confounding because of other factors. However, we attempted to recruit an appropriate comparison group in the non-exposed controls by excluding medical staff, including physicians and nurses, at the clinic to reduce possible residual confounding factors. Finally, our sample was not selected randomly from the corresponding areas or groups, which may have led to selection bias in the study. However, a relatively small number of CMD patients were alive at the time of the study and we did not select the moderately exposed residents based on their neurological or neurocognitive deficits; we, thus, believe that the current sampling method was reasonable and realistic. 

## 5. Conclusions

Since the first report of Minamata disease, only “severe” cases of in utero exposure (i.e., CMD patients) have been highlighted in relation to the disease. However, this study suggests that residents who experienced low-to-moderate methylmercury exposure in utero, lower than that of CMD patients, also have neurological or neurocognitive impairments, probably due to diffuse damage to the brain. These deficits may have a negative impact on daily life at school, at home, and at work. Thus, more attention should be paid to mild forms of CMD; that is, to individuals who had low-to-moderate in utero methylmercury exposure in the Minamata area.

## Figures and Tables

**Figure 1 ijerph-20-06173-f001:**
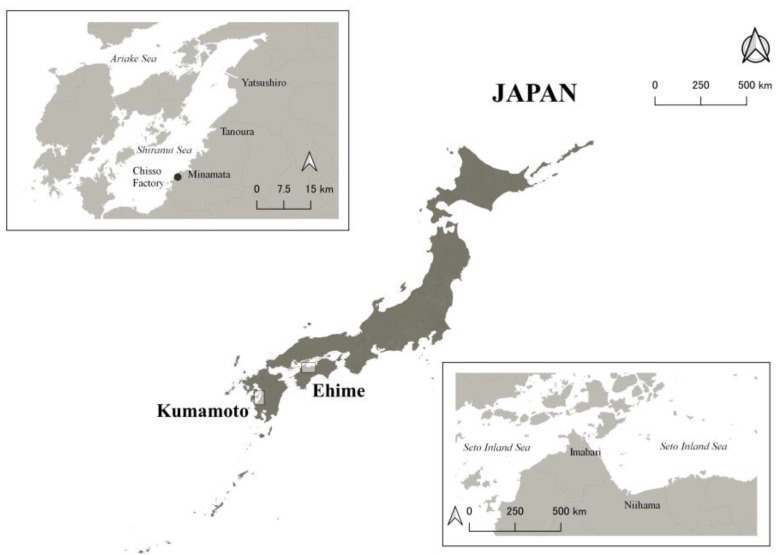
A map of the study areas. The upper left rectangle shows Minamata City and surrounding areas, and the lower right rectangle shows Niihama City and surrounding areas.

**Table 1 ijerph-20-06173-t001:** Demographic characteristics of study participants grouped by exposure status (*n* = 52).

	Non-Exposed Residents	Moderately Exposed Residents	Congenital Minamata Disease Patients
	(*n* = 27)	(*n* = 15)	(*n* = 10)
Mean age, years (SD)	59.9 (3.0)	62.2 (3.9)	63.3 (3.5)
Male, *n* (%)	8 (29.6)	6 (40.0)	7 (70.0)
Drinker, *n* (%)	12 (44.4)	9 (60.0)	4 (40.0)
Smoking, *n* (%)			
Never smoked	16 (59.3)	10 (66.7)	7 (70.0)
Ex-smoker	6 (22.2)	4 (26.7)	3 (30.0)
Current smoker	5 (18.5)	1 (6.7)	0 (0)
Educational attainment, *n* (%)			
Junior high school	0 (0)	5 (33.3)	7 (70.0)
High school	13 (48.2)	9 (60.0)	2 (20.0)
Higher than high school	14 (51.9)	1 (6.7)	1 (10.0)
Self-rated health, *n* (%)			
Good/very good	18 (66.7)	1 (6.7)	2 (20.0)
Moderate	8 (29.6)	5 (33.3)	5 (50.0)
Bad/very bad	1 (3.7)	9 (60.0)	3 (30.0)

SD: standard deviation.

**Table 2 ijerph-20-06173-t002:** Distributions (mean (SD)) of the neuropsychological tests grouped by exposure status.

	Non-Exposed Residents	Moderately Exposed Residents	Congenital Minamata Disease Patients	*p*-Values ^a^
MOCA-J	28 (1.5)	23.1 (5.2)	9.6 (6.4)	<0.001
WAIS-III				
Digit symbol coding	12.9 (2.4)	7.6 (3.2)	NA ^b^	<0.001
Block design	11.9 (2.6)	9.1 (2.8)	4 (4.2) ^c^	<0.001
Information	10.3 (2.4)	7.9 (2.9)	3.8 (1.5) ^c^	<0.001
Symbol search	13 (4.6)	8.7 (3.9)	NA ^b^	0.004
Keio version of the Wisconsin Card Sorting Test				
Categories achieved	4.1 (2)	1.9 (2)	NA ^b^	0.001
Perseverative errors of Nelson	4.1 (9.7)	9.1 (7.7)	NA ^b^	0.089
Total errors	17.4 (10.1)	28.9 (11.4)	NA ^b^	0.002
Non-perseverative errors of Nelson	13.7 (7.1)	19.8 (6.1)	NA ^b^	0.008
Grooved Pegboard Test				
Dominant hand	63.1 (8.9)	101 (57.9)	173 (58) ^d^	<0.001
Non-dominant hand	67.8 (10.5)	117.1 (93.3)	240.8 (72.3) ^d^	<0.001

MOCA-J: Japanese version of the Montreal Cognitive Assessment; NA: not available; SD: standard deviation; WAIS-III: Wechsler Adult Intelligence Scale III. ^a^ Obtained from the analysis of variance for three-group comparisons or the t-test for two-group comparisons. ^b^ Unavailable because congenital Minamata disease patients were unable to perform the tests. ^c^ Several patients were unable to perform the block design (three patients) and information (one patient) subtests. ^d^ Several patients were unable to perform or complete the test with the dominant (seven patients) or non-dominant (six patients) hand.

**Table 3 ijerph-20-06173-t003:** Adjusted ^a^ beta coefficients (with their 95% CIs) for the associations between exposure status and neuropsychological tests.

	Non-Exposed Residents	Moderately Exposed Residents	Congenital Minamata Disease Patients
MOCA-J	reference	−4.11 (−6.78, −1.43)	−16.77 (−20.08, −13.46)
WAIS-III			
Digit symbol coding	reference	−5.11 (−6.84, −3.37)	NA ^b^
Block design	reference	−2.32 (−4.27, −0.36)	−6.78 (−9.48, −4.09)
Information	reference	−2.41 (−4.10, −0.73)	−6.66 (−8.76, −4.55)
Symbol search	reference	−4.60 (−7.51, −1.70)	NA ^b^
Keio version of the Wisconsin Card Sorting Test			
Categories achieved	reference	−1.91 (−3.25, −0.56)	NA ^b^
Perseverative errors of Nelson	reference	4.28 (−2.08, 10.64)	NA ^b^
Total errors	reference	10.51 (3.06, 17.96)	NA ^b^
Non-perseverative errors of Nelson	reference	5.64 (0.87, 10.40)	NA ^b^
Grooved Pegboard Test			
Dominant hand	reference	39.14 (14.49, 63.79)	100.86 (52.55, 149.16)
Non-dominant hand	reference	58.84 (21.68, 96.00)	181.58 (118.01, 245.15)

CI: confidence interval; MOCA-J: Japanese version of the Montreal Cognitive Assessment; NA: not available; WAIS-III: Wechsler Adult Intelligence Scale III. ^a^ Adjusted for age and sex. ^b^ Unavailable because congenital Minamata disease patients were unable to perform the tests.

## Data Availability

The data presented in this study are available on request from the corresponding author. The data are not publicly available due to ethical reasons.

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
