# Peer review of "Neurological and Neurocognitive Impairments in Adults with a History of Prenatal Methylmercury Poisoning: Minamata Disease"

_ijerph, 2023, doi:10.3390/ijerph20126173_

Round 1

Reviewer 1 Report

The manuscript ' Neurological and neurocognitive impairments in adults with a  history of prenatal methylmercury poisoning: Minamata disease" by Yorifuji et al . I have some questions  about the evaluation of  Neuropsychological tests: 

1. This test only establishes an association between CMD patients, moderately exposed residents, and non-exposed controls , but doesn't say   about  non-demented elderly persons and in those at genetic risk for neurodegenerative  diseases

2. Many factors have potential to affect  cognitive impairment ?

3.  The study says nothing about how much mother and child were contaminated with Hg

4.  Would the duration of the problem not affect the WCST score?

Minor editing of English language required

Author Response

Responses to comments from reviewer #1

Thank you very much for your valuable comments to clarify the manuscript. Our responses to your comments are described below in a normal font following your comments in boldface.

The manuscript 'Neurological and neurocognitive impairments in adults with a history of prenatal methylmercury poisoning: Minamata disease" by Yorifuji et al. I have some questions about the evaluation of Neuropsychological tests:

  1. This test only establishes an association between CMD patients, moderately exposed residents, and non-exposed controls, but doesn't say about non-demented elderly persons and in those at genetic risk for neurodegenerative diseases.

Reply:

Thank you for your comment. As you pointed out, there may be residual confounding because of other factors. We thus noted it as one of the limitations in the Discussion section as follows (lines 295-296):

“Also, there is a possibility of residual confounding because of other factors.”

  1. Many factors have potential to affect cognitive impairment?

Reply:

Thank you for your comment. This is also related with the response to the first question. We note this as follows (lines 291-296):

“Third, we did not adjust for educational attainment or self-rated health because these variables were considered to be intermediate variables between methylmercury exposure and outcomes, and we were unable to adjust for the socioeconomic status or intelligence quotient of the participants’ parents, which might confound any associations. Also, there is a possibility of residual confounding because of other factors.”

  1. The study says nothing about how much mother and child were contaminated with Hg.

Reply:

We appreciate your comment to clarify the point. We already describe the level of the exposure among the participants as follows (lines 197-200):

“Umbilical cord methylmercury concentrations were available from two CMD patients and seven moderately exposed participants; the average concentrations were 1.67 parts per million (ppm) and 0.77 ppm, respectively.”

  1. Would the duration of the problem not affect the WCST score?

Reply:

Thank you for your comment. We are not sure whether the duration of Minamata disease problem can affect the findings from the neuropsychological tests. However, the long-term process of the history was caused by the incident; therefore, even if the duration could affect the neuropsychological findings, it would be the effect of the poisoning.

Author Response

Responses to comments from reviewer #2

Thank you very much for your valuable comments to clarify the manuscript. Our responses to your comments are described below in a normal font following your comments in boldface.

“Neurological and neurocognitive impairments in adults with a history of prenatal methylmercury poisoning: Minamata disease.” This study has a small sample size. However, study results suggest important insights into the relationships between prenatal methylmercury exposures and neurobehavioral dysfunctions in Minamata poisoning. These findings contribute to the understanding of methylmercury poisoning and carry significant implications for environmental health science. However, this manuscript requires major revisions in the methods, results, and discussion sections.

Abstract:

If the Japanese government or another study has monitored exposed doses of methylmercury exposure in mothers or children for both CMD and moderately exposed groups, please add their exposed dose. Even estimated concentrations of methylmercury exposures would help to understand how much dosage they were. I suspect the moderately exposed group would be considered as having high exposure, compared to the current EPA guidelines. The explanations can be included in the method section. In the abstract, authors can add the exposed rage or mean if it is possible.

Reply:

Thank you for your helpful comments. Following your comments, we include the exposure information of the participants as follows (lines 21-23):

“The average umbilical cord methylmercury concentrations were 1.67 parts per million (ppm) for CMD patients and 0.77 ppm for moderately exposed participants.”

Line 24, “ although the damage was more severe in the CMD patients.” Please avoid using casual language. You can use “score decline” or “score decrements” instead of “the damage”.

Reply:

Following your comment, we modified it as follows (lines 25-26):

“, although the score decline was more severe in the CMD patients.”

Line 25 - 26, 95% confidence interval can change be changed 95% CI.

Reply:

Following your comment, we modified them.

Introduction:

Please add a short summary of Minamata poisoning. Minamata disease is famous, but some researchers do not know well about historical methylmercury poisoning. Why Minamata disease occurred? What company released methylmercury and mercury catalyzer into Minamata Bay? How long that company had released methylmercury?

Reply:

Thank you for your comment to clarify the manuscript. Following your comment, we added a short summary of Minamata disease as follows (lines 36-42):

“Minamata disease, which occurred in Minamata in the 1950s and 1960s, is well known as a food poisoning caused by fish contaminated with methylmercury [1, 2]. The first patient was officially notified to the local Public Health Center in May 1, 1956. The source and transmission mode were contaminated fish and shellfish. The etiologic agent, methylmercury, was discharged from a chemical factory into Minamata Bay and the Shiranui Sea from 1932 until 1968; no effective preventive measures were taken during this period [3, 4].”

From line 55 - 62, what was the age range of participating adolescents in epidemiological studies? What were their estimated methylmercury exposures?

Reply:

Thank you for your comment. Although we do not have the estimated methylmercury exposure for the participants, we added the age range of the participants as follows (lines 62-64):

“Furthermore, Fujino et al. conducted a cross-sectional study in 1970 to assess the neuro-logical status of adolescents aged 12 to 15 years born between 1955 and 1958 in the ex-posed areas;”

Methods

Line 77-78, what are the criteria for CMD or certified Minamata disease according to the Japanese government? Are these criteria only including symptoms instead of methylmercury-exposed levels?

Reply:

Thank you for your useful comment. To be certified, residents need a combination of neurological signs as well as an exposure history and one of the requirements for the exposure history mentions the concentration of methylmercury in umbilical cords. We thus introduce the following information (lines 86-89):

“To be certified, residents should have a combination of neurological signs with an exposure history and one of the requirements for the exposure history for the CMD patients include that the patients should have umbilical methylmercury concentration above 1ppm [15].”

Line 79 – 86, please clarify the participant’s grouping. If authors want to name the moderately exposed groups, exposed methylmercury concentrations need to be explained. If participants were grouped with cognitive symptoms or Japanese government criteria, the groups would be the severe cognitive dysfunction (CMD), non-severe cognitive dysfunction (no-CMD), and non-exposed group. Heavy

methylmercury exposures result in cognitive dysfunction. Therefore, if the authors estimated participants were moderately exposed to methylmercury and categorized into the moderately exposed group, it needs to be explained.

Reply:

Thank you for your helpful comment. We agree with your concern. Following your comment, we introduced the exposure information of the participants in the abstract. Also, we already noted this in the Results section. Moreover, we introduce the exposure information in the previous studies conducted outside of Japan in the Discussion section. We hope these modifications can help readers to understand the exposure status and grouping of our participants.

Line 86 – 88, “ The etiologic… .. this period.”, these sentences need to move to the introduction. Please read my suggestion above.

Reply:

Following your comment, we moved the sentence to the short summary of Minamata disease in the Introduction section.

Results

Line 185-188, this sentence is important. This sentence is necessary for readers to understand each group's exposure level. Please read my suggestion in the Method.

Reply:

Thank you for your helpful comment to clarify the manuscript. As mentioned, we modified the manuscript following your advice.

Table 2, 3 and 4, can be summarized into two tables, the mean scores (SD) table and beta coefficients (95% CI) table. The results from score distributions (means or medians) and multiple linear regressions were different meaning, distributions or model fit. With the mean score and beta coefficients tables, authors can perform pair-wise tests or ANOVA to add p-values to the moderately exposed and CMD groups, and add p-values to the beta coefficients.

Reply:

Thank you for your comment. Following your comment, we summarized the previous 3 table to 2 tables. We introduce the score distributions in new Table 2 and the beta coefficients in new Table 3, but we did not add p-values for the beta coefficients because we already introduce 95% CIs for them and both p-values and 95%CIs can provide similar meanings.

Line 196, 95% confidence interval can be 95% CI.

Reply:

Following your comment, we modified them.

Discussion

Line 214, please avoid using casual language for results from this study. For example, effects, caused, damage etc. Authors can replace them with associations, relationships, or suggest.

Reply:

Following your comment, we avoided using causal language in the Discussion section.

Line 216 – 219, a short summary of the previous study can be added here, 2 or 3 sentences.

Reply:

Thank you for your comment to clarify the point. We introduce a short summary of the previous findings on the CMD patients as follows (lines 232-233):

“Indeed, a previous review noted a high prevalence of intellectual disability among CMD patients [1].”

Line 224 – 230, this paragraph needs to add more information. What is the study design? Is that an animal study, epidemiological study, or systematic review? How were participants exposed to methylmercury? What is exposed dosage? For example, “A recent systematic review report that xxxxxx.”

Reply:

Thank you for your comment to clarify the content. These are systematic reviews to mention the developmental neurotoxicity of methylmercury. Following your comment, we introduce one of the reviews we cited in a detail. We hope this modification helps readers to understand how the participants in the previous studies were exposed, the exposure range of the participants in the previous studies, and etc. We added the following text into the paragraph (lines 241-252):

“For example, a systematic review included 27 epidemiologic studies and examined possible developmental neurotoxicity of methylmercury [26]. Most of the included studies were based on communities with high fish intake, while a small number of studies were based on general populations. Even the residents in the communities with high fish intake are considered to have lower methylmercury concentrations (e.g., average maternal hair mercury concentrations < 10 ppm) than the exposed population in Minamata (e.g., median hair mercury concentration among healthy fisherman in Minamata was 30 ppm in 1960 [6]). Then, the review noted that most of the studies found neurodevelopmental impairments because of prenatal methylmercury exposure and the affected neurocognitive domains included areas related with as attention, language, motor, and visuospatial functions [26]; the current findings support these previous results.”

Line 242 – 251, please clarify this paragraph. What is the difference in exposed methylmercury concentrations between localized and diffused damage in Minamata?

Reply:

Thank you for your comment to clarify the point. We do not have information on the difference in exposure concentrations between localized and diffused damage in Minamata, but the difference in the damaged areas would be more related with timing of the exposure. We thus amended the sentence as follows (lines 264-265):

“Methylmercury is neurotoxic for the central nervous system and considered to affect different regions of the brain depending on the exposure timing.”

Line 255 – 257, please add more information. What kind of studies are.

Reply:

Thank you for your useful comments. We think the above-mentioned systematic review would be more helpful for readers to understand the content, we thus replaced the references with the systematic review and modified the sentences as follows (lines 277-299):

“As mentioned, on the basis of previous studies included in the systematic review [26], this level is higher than the value that is thought to induce neurocognitive impairments.”
